# DPIC: Decoupling Prompt and Intrinsic Characteristics for LLM Generated Text Detection

**Xiao Yu**[1,2†]**, Yuang Qi**[1,2†]**, Kejiang Chen**[1,2∗]**, Guoqiang Chen**[1]
**Xi Yang**[1]**, Pengyuan Zhu**[3]**, Xiuwei Shang**[1]**, Weiming Zhang**[1]**, Nenghai Yu**[1]

[1]University of Science and Technology of China, China
[2]Key Laboratory of Cyberspace Security, Ministry of Education, China
[3]Hefei High-dimensional Data Technology, China

`yuxiao1217@mail.ustc.edu.cn, {chenkj, zhangwm, ynh}@ustc.edu.cn`

## Abstract

Large language models (LLMs) have the potential to generate texts that pose risks of misuse, such as plagiarism, planting fake reviews on e-commerce platforms, or creating inflammatory false tweets. Consequently, detecting whether a text is generated by LLMs has become increasingly important. Existing high-quality detection methods usually require access to the interior of the model to extract the intrinsic characteristics. However, since we do not have access to the interior of the black-box model, we must resort to surrogate models, which impacts detection quality. In order to achieve high-quality detection of black-box models, we would like to extract deep intrinsic characteristics of the black-box model generated texts. We view the generation process as a coupled process of prompt and intrinsic characteristics of the generative model. Based on this insight, we propose to decouple prompt and intrinsic characteristics (DPIC) for LLM-generated text detection method. Specifically, given a candidate text, DPIC employs an auxiliary LLM to reconstruct the prompt corresponding to the candidate text, then uses the prompt to regenerate text by the auxiliary LLM, which makes the candidate text and the regenerated text align with their prompts, respectively. Then, the similarity between the candidate text and the regenerated text is used as a detection feature, thus eliminating the prompt in the detection process, which allows the detector to focus on the intrinsic characteristics of the generative model. Compared to the baselines, DPIC has achieved an average improvement of 6.76% and 2.91% in detecting texts from different domains generated by GPT-4 and Claude3, respectively.

## 1 Introduction

Large language models (LLMs) such as PaLM [6], ChatGPT [29], LLaMA [38], and GPT-4 [1] exhibit advanced language capabilities for understanding natural language and solving complex tasks via text generation. The outstanding performance of LLMs has led to the belief that they can be the artificial general intelligence (AGI) of this era [5]. However, if placed in the wrong hands, LLMs such as ChatGPT can undoubtedly serve as a "weapon of mass deception" [35]. For example, the formidable writing capabilities of ChatGPT pose a significant threat to democracy, as they enable the creation of automated bots on online social networks that can manipulate people's political choices during election campaigns [36, 13]. Furthermore, the adoption of ChatGPT by students in educational institutions has led to instances of academic dishonesty, with essays and assignments being generated

---

∗Corresponding author.
†Equal contribution.

38th Conference on Neural Information Processing Systems (NeurIPS 2024).

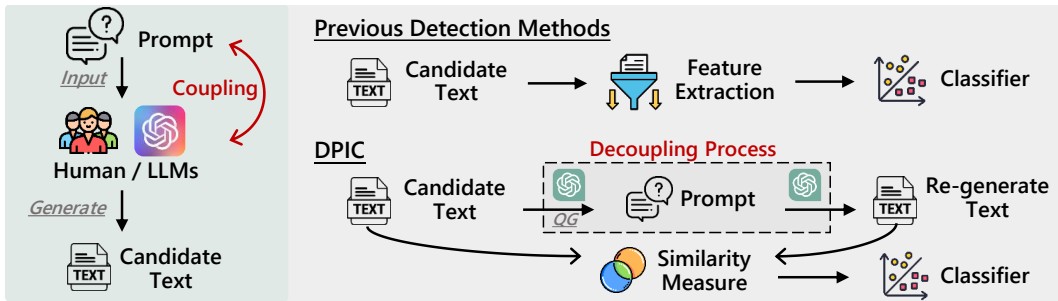

Figure 1: The distinctions between DPIC and previous detection methods. DPIC extracts the deep intrinsic characteristics of the black-box model generated texts by decoupling the prompt and the intrinsic characteristics of the generative model.

through its use, as reported by various news sources [23, 31]. Therefore, it is essential and urgent to detect LLM-generated texts.

Existing detectors can be grouped into two main categories: supervised classifiers [10, 25, 32, 41, 21] and zero-shot classifiers [12, 24, 37, 42, 4, 15]. Supervised classifiers are usually trained to distinguish between LLM-generated texts and human-created texts. Supervised classifiers achieve strong detection performance in detecting datasets belonging to the same domain as the training set, but usually fail when faced with datasets that are not in the domain of the training set. [32, 3, 39]. As a result, researchers have shifted their perspective to zero-shot classifiers. The zero-shot classifiers directly use pre-trained language models without fine-tuning to gather statistical features and are immune to domain-specific degradation. In general, zero-shot classifiers exhibit better generalizability. However, zero-shot classifiers require model knowledge to extract intrinsic characteristics and perform reliable detection, which is unrealistic for black-box models. Most powerful LLMs, such as ChatGPT, are close-source. As a result, zero-shot classifiers cannot achieve the desired quality of detection when detecting texts generated by closed-source models [2, 29, 1]. To achieve a high quality of detection of black-box models, we would like to extract the deep intrinsic characteristics of the black-box models. In this case, we can only rely on the text inputs and outputs for black-box models. The remaining and significant challenge is how to extract deep intrinsic characteristics of the black-box model generated texts.

In our view, *the generation process can be seen as a coupled process of prompt and intrinsic characteristics of the generative model.* The texts generated by LLMs and created by humans exhibit inconsistent intrinsic characteristics, central to distinguishing the two. *The discriminative features should rely more on the intrinsic characteristics to achieve a general detection.* Zero-shot detectors extract intrinsic characteristics by relying on internal information of the generative model to identify texts generated by LLMs [4], achieving desired detection performance on white-box models. However, when confronted with black-box models, these zero-shot detectors do not have access to the model's internal information and must rely on open-source surrogate models, leading to detection performance degradation. Supervised detectors utilize only text for detection and do not require access to the model internals. To eliminate the interference of prompts, the supervised detectors often require large-scale datasets generated with various prompts and then focus on the intrinsic characteristics of the generative model. However, it is impractical to include all prompts in the dataset because of the diversity of prompts. This leads to the question of *how to extract the intrinsic characteristics from the black-box model generated texts skillfully?*

In this paper, we propose to **d**ecouple **p**rompt and **i**ntrinsic **c**haracteristics (**DPIC**) to extract deep intrinsic characteristics from the black-box model generated texts for detection. Specifically, given a candidate text, we utilize an auxiliary LLM, leveraging the LLM's powerful inductive capabilities, to reconstruct the prompt based on the candidate text. The reconstructed prompt is then used for the auxiliary LLM to obtain the regenerated text. This process aims to make the candidate and regenerated texts align with their prompts, respectively. Then, by comparing the similarity between the candidate text and the regenerated text, we can determine whether the candidate text is generated by LLMs or created by humans. In this case, the candidate text and the regenerated text are compared with aligned prompts, which allows the detector to focus on the deep intrinsic characteristics. With the decoupling process, we extract the deep intrinsic characteristics of black-box model generated

texts by decoupling the prompt and the intrinsic characteristics. This provides the detector with richer information about deep intrinsic characteristics and thus achieves higher detection quality and better generalizability. Figure 1 shows the distinctions between our proposed detection method and previous detection methods.

The main contributions of our work are as follows:

- **New insight**. We reconsider the generated text detection methods and propose a new perspective to improve generalizability while ensuring detection quality. In our view, prompt and intrinsic characteristics of the generative model in generated text are tightly coupled together, which limits the generalizability of the detector. During the detection process, the detector should rely more on the intrinsic characteristics to achieve better generalizability.
- **Ingenious approach**. we propose to decouple prompt and intrinsic characteristics (DPIC) to extract the deep intrinsic characteristics of the black-box model generated texts. In this process, the candidate text and the regenerated text are aligned with their respective prompts. We then use the similarity between the candidate text and the regenerated text as discriminative features, reducing the impact of prompts on the detection process and allowing the detector to focus more on intrinsic characteristics.
- **Impressive performance**. DPIC achieves 6.76% and 2.91% average improvement in detection performance compared to the baselines in detecting texts from different domains generated by two commercial closed-source models: GPT-4 and Claude3. These findings underscore the efficacy of our proposed method.

## 2 Related work

LLM-generated text detection could increase trust in natural language generation systems and encourage adoption. Given its significance, there has been a growing interest in academia and industry to pursue research on LLM-generated text detection. Current LLM-generated text detection is categorized into supervised [36, 25] and zero-shot methods [12, 24, 37, 42, 4].

Previous research focuses on supervised methods, where classifiers are trained to differentiate between texts generated by LLMs and texts created by humans. For example, OpenAI [30] trained the OpenAI text classifier on a collection of millions of texts. RADAR [16] introduces the idea of adversarial learning to train a detector that can resist paraphrase attacks. However, supervised methods exhibit shortcomings in terms of generalization. Therefore, to obtain more generalizable detection methods, current researchers work on developing zero-shot methods. Existing zero-shot methods primarily rely on statistical features, leveraging pre-trained large language models to gather them. In our view, these methods can be seen as extracting the intrinsic characteristics of LLMs in different ways. For example, some researchs [36, 17] take advantage of the rank or entropy of each word in a text conditioned on the previous context to represent the intrinsic characteristics. The other methods represent the intrinsic characteristics by different features, including average probability, and top-K buckets [12], likelihood [15], probability curvature [24], divergence between multiple completions of a truncated passage (DNA-GPT) [42], and conditional probability curvature [4].

Due to excessive reliance on training data, the generalizability of supervised methods is questionable [32, 3, 39]. That is, the reliability and robustness of the detection methods may significantly decrease when applied to out-of-domain data. In order to achieve generalizability, existing researchers have worked on developing zero-shot detection methods. Zero-shot detection methods are immune to domain-specific degradation and are on par with supervised classifiers in terms of detection performance. However, from the experimental results, the detection performance of zero-shot detectors for black-box models will be significantly reduced compared to white-box models. Therefore, the primary existing challenge is developing detection methods with high detection performance and high generalizability for black-box models.

## 3 Threat models

Given a candidate text $x$, which may be human-created or generated by a language model, the detector is dedicated to distinguishing between these two sources. In the white-box scenario, the detector has the advantage of accessing the potential source model, including weights and probability distributions.

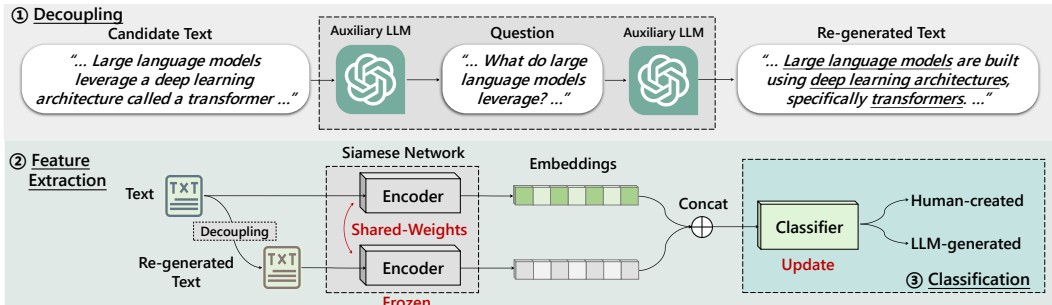

Figure 2: An overview of DPIC. Given a candidate text, we utilize an auxiliary LLM to reconstruct the prompt based on the candidate text. The reconstructed prompt is then used for the auxiliary LLM to obtain the regenerated text. This process aims to make the candidate and regenerated texts align with their prompts, respectively. Then, by comparing the similarity between the candidate text and the regenerated text, we can determine whether the candidate text is generated by LLMs or created by humans.

Conversely, in the black-box scenario, only the text input and output of the potential source model are accessible. In this paper, we focus on the black-box scenario.

# 4 Method

## 4.1 Motivation

Exploiting the intrinsic characteristics of the text generated by LLMs is an effective way to improve the detection quality and generalizability of the detector. DetectGPT [24] and Fast-DetectGPT [4] can be regarded as adopting log probabilities as the intrinsic characteristic of the LLM. However, the log probabilities cannot be accessible when facing black-box scenarios involving closed-source models such as GPT-4 and Claude3.

In fact, the intrinsic characteristics of the model are not only reflected in the logistic probabilities. The generated text itself is a coupling of the prompt and intrinsic characteristics. However, extracting the intrinsic characteristics from the generated text is challenging because the prompts are diverse, covering various topics, and are decoupled with the inherent characteristics. In supervised detection methods, it is very important to extend the dataset as much as possible so that the detector is familiar with the topic of the text to be detected. However, it is not feasible for the model to memorize an infinite number of text topics. We aim to abandon the need for the detector to perform topic recognition. Instead, we hope to provide the topic information of the texts to be detected to the detector each time it performs a detection. The requirement reminds us of the powerful capabilities of LLMs, which can regenerate text through induction and generation. In the comparison perspective between the text to be detected and the regenerated text, the detection bias caused by the text topics can be alleviated.

Based on the above analysis, we propose a supervised detection method based on decoupling prompt and intrinsic characteristics (DPIC), as illustrated in Figure 2, including three modules: decoupling process, feature extraction, and classification.

## 4.2 Decoupling process

Given a candidate text $\mathbf{x}$ and an auxiliary LLM $\mathcal{M}_{\text{aux}}$, the decoupling process aims to to obtain a regenerated text $\hat{\mathbf{x}}$ that is consistent with $\mathbf{x}$ under the semantic constraints of the prompt $\mathbf{p}$, i.e., $\hat{\mathbf{x}} \leftarrow \arg\max_{\hat{\mathbf{x}}} P_{\mathcal{M}_{\text{aux}}}(\hat{\mathbf{x}}|\mathbf{p})$, where $\leftarrow$ represents the sampling process of LLMs from the predicted distribution. However, since the prompt used to generate $\mathbf{x}$ is unknown to the detector in the black-box setting, we first need to reconstruct the prompt $\mathbf{p}$ of $\mathbf{x}$. When the log probabilities are hidden, the recently proposed prompt recovery method [27] based on inverting probabilities is inapplicable. We, therefore, shift our focus to another similar task, namely question generation (QG) [8, 9, 20]. QG involves generating natural-sounding questions based on given sentences or paragraphs. The QG task

can be defined as finding $\bar{\mathbf{q}}$, such that:

$$\bar{\mathbf{q}} \leftarrow \arg\max_{\mathbf{q}} P(\mathbf{q}|\mathbf{t}), \tag{1}$$

where $P(\mathbf{q}|\mathbf{t})$, is the conditional log-likelihood of the predicted question sequence $\mathbf{q}$, given the input $\mathbf{t}$. We notice that the LLM performs exceptionally well in the QG task. So we directly utilize $\mathcal{M}_{\text{aux}}$ to reconstruct the prompt.

$$\hat{\mathbf{p}} \leftarrow \arg\max_{\hat{\mathbf{x}}} P_{\mathcal{M}_{\text{aux}}}(p_{rc}\|\mathbf{x}), \tag{2}$$

where $p_{rc}$ represents the prompt template used to query $\mathcal{M}_{\text{aux}}$ reconstructing the prompt. After obtaining the reconstructed prompt $\hat{\mathbf{p}}$ of $\mathbf{x}$, the next step is to utilize $\mathcal{M}_{\text{aux}}$ to get $\hat{\mathbf{x}}$ based on $\hat{\mathbf{p}}$.

$$\hat{\mathbf{x}} \leftarrow \arg\max_{\hat{\mathbf{x}}} P_{\mathcal{M}_{\text{aux}}}(\hat{\mathbf{p}}\|p_{rg}), \tag{3}$$

where $p_{rg}$ regenerates the regeneration template we use to query $\mathcal{M}_{\text{aux}}$. The details of $p_{rc}$ and $p_{rg}$ are shown in Appendix B. In the aforementioned process, $\mathcal{M}_{\text{aux}}$ can be a black box to the user, where the user only inputs the text to be paraphrased or regenerated, and obtains the corresponding regenerated text. The decoupling process can be formally represented as:

$$\hat{\mathbf{x}} \leftarrow \mathcal{M}_{\text{aux}}(\mathbf{x}, p_{rc}, p_{rg}). \tag{4}$$

### 4.3 Feature extraction and classification

Through the decoupling process, we get $\hat{\mathbf{x}}$ aligned with $\mathbf{x}$ on the reconstructed prompt $\hat{\mathbf{p}}$. After that, we measure the similarity of $\mathbf{x}$ and $\hat{\mathbf{x}}$, then use it as a discriminative feature. Since the two texts are consistent at the prompt level, the similarity only indicates the intrinsic characteristics contained in $\mathbf{x}$. The hypothesis under our detection method is that the text generated by LLMs and text created by humans exhibit inconsistent intrinsic characteristics. If the candidate text is generated by LLMs, the regenerated text will exhibit higher similarity to the candidate text because the texts generated by LLMs have similar intrinsic characteristics. On the contrary, if the candidate text is created by humans, then the similarity will be lower because the text generated by LLMs and text created by humans exhibit intrinsic characteristics.

We employed a Siamese Network to measure the similarity between texts. A Siamese Network is a neural network that simultaneously processes two different input tensors using the same weights, producing comparable output tensors. $\mathbf{x}$ and $\hat{\mathbf{x}}$ separately enter two subnets that share structures, parameters, and weights. For each subnet, we employ a language model as an encoder to perform feature extraction on the texts, which can be formulated as below.

$$\mathbf{v}_{\mathbf{x}} = \mathcal{E}(\mathbf{x}), \mathbf{v}_{\hat{\mathbf{x}}} = \mathcal{E}(\hat{\mathbf{x}}), \tag{5}$$

where $\mathcal{E}$ donates the encoder, $\mathbf{v} \in \mathbb{R}^d$. Finally, $\mathbf{v}_{\mathbf{x}}$ and $\mathbf{v}_{\hat{\mathbf{x}}}$ are concatenated and fed into a classifier denoted as $f : \mathbb{R}^{2\times d} \to \mathbb{R}^2$. Specifically, we use `gte-Qwen1.5-7B-instruct`[†] as the encoder which can encode texts with a maximum of $32K$ tokens into embeddings of 4096 dimensions, while the classifier consists of three fully connected layers with ReLU function. The dimensions of the intermediate layers in the classifier are $1024$ and $512$, respectively.

We freeze the weights of the Siamese encoder and train the classifier using binary cross-entropy loss. The loss function can be formalized as:

$$\mathcal{L} = -\frac{1}{N}\sum_{i=1}^{N}\left(y^{(i)}\log(y_{\text{pred}}^{(i)}) + (1-y^{(i)})\log(1-y_{\text{pred}}^{(i)})\right), \tag{6}$$

where $y$ represents the true label of $\mathbf{x}$, and the predicted label of the detector is represented as $y_{\text{pred}} = f\left(\mathcal{E}\left(\mathbf{x}\right)\|\mathcal{E}\left(\mathcal{M}_{\text{aux}}\left(\mathbf{x}, p_{rc}, p_{rg}\right)\right)\right)$.

## 5 Experiment

### 5.1 Implementation details

**Auxiliary models.** We select two auxiliary models for the decoupling process, including ChatGPT[‡] and Vicuna-7b-v1.5[§]. For ChatGPT, we directly use the ChatGPT API for querying. If Vicuna is

---

[†]https://huggingface.co/Alibaba-NLP/gte-Qwen1.5-7B-instruct
[‡]https://platform.openai.com/docs/models/gpt-3-5-turbo
[§]https://huggingface.co/lmsys/Vicuna-7b-v1.5

chosen as the auxiliary model, we locally deploy a Vicuna-7b model but treat it as a black-box, solely obtaining its output responses without accessing the model parameters or probability distributions.

**Training.** Since the DPIC method we propose is supervised, it necessitates the selection of a training set. In this paper, we used the open-source Human-ChatGPT Comparison Corpus (HC3) [14] dataset collected by previous researchers as a training set to ensure the reproducibility of our approach. HC3 dataset consists of both human and ChatGPT responses to the same prompts. When the auxiliary model is set to ChatGPT, we select these responses from the HC3 dataset as the texts to be detected. When the intermediary model is Vicuna, we replace the texts generated by ChatGPT in the HC3 dataset with the texts generated by Vicuna as the machine-generated texts in the training set.

**Testing.** We evaluated the performance of the detection methods on texts generated by three current widely used commercial closed-source models, including ChatGPT (gpt-3-5-turbo) [¶], GPT-4 (gpt-4-0613) [‖], and Claude3 (claude-3-opus-20240229) [**]. We refer to these models as the source models for LLM-generated texts.

We use three datasets that are not part of the domains covered in the HC3 dataset to evaluate the generalizability of detection methods fully. The three datasets are Xsum [28] for news articles, WritingPrompts [11] for story writing, PubMedQA [18] for biomedical research question answering, which are consistent with previous work [4] in the field. We use the three different source models to generate texts on the aforementioned datasets. The original texts from the datasets and the texts generated by the models together serve as the data to be detected in the testing process.

We also created other datasets for three different domains, and evaluated the detection performance of our methods and baselines on these datasets. The details of these datasets and the results are displayed in Appendix D.

**Evaluation metric.** We measure the detection performance in the area under the receiver operating characteristic (AUROC). AUROC ranges from 0.0 to 1.0, mathematically denoting the probability of a random machine-generated text having a higher predicted probability of being machine-generated than a random human-written text. A higher AUROC value indicates a better detection quality.

**Baselines.** We compared DPIC with existing supervised detectors and zero-shot detectors. For supervised detectors, we compared GPT-2 detectors based on RoBERTa-base/large [22] crafted by OpenAI and RADAR [16]. For zero-shot detectors, we selected DNA-GPT [42], DetectGPT [24], and its enhanced variants NPR [37] and Fast-DetectGPT [4]. We also chose classic zero-shot classifiers, including Likelihood (mean log probabilities)[12], LogRank (average log of ranks in descending order by probabilities) [36], Entropy (mean token entropy of the predictive distribution)[17], and LRR (an amalgamation of log probability and log-rank)[37].

### 5.2 Performance

**Detection effectiveness.** The detection performance of DPIC and baselines is shown in Table 1. When using ChatGPT as the auxiliary LLM, our method achieves an average AUROC of 96.34%, 97.34%, and 98.78%, respectively, in detecting ChatGPT, GPT-4, and claude3. By comparison, Fast-DetectGPT, which achieves the highest AUROC among open source baselines, has lower AUROC of 96.15%, 90.58%, and 95.87%, respectively. It can be seen that our method outperforms the baselines in terms of average detection quality, particularly in detecting advanced commercial closed-source models like GPT-4 and Claude3, with a significant advantage. Meanwhile, when using Vicuna-7b as the auxiliary model, our method achieves an average AUROC of 95.58%, 96.74%, and 98.75% in detecting ChatGPT, GPT-4, and claude3, respectively. Although our training set only includes LLM-generated text from either ChatGPT or Vicuna-7b, DPIC achieves high detection AUROC for text originating from three different source models when using the two auxiliary models. This indicates that our method does not require prior knowledge of the text's source model, as the differences in features between LLM-generated and human-created text are greater than the differences among texts generated by various models. Besides, the effectiveness of the open-source auxiliary model demonstrates that DPIC can conduct detection at a relatively low cost, indicating the practical efficacy of our detection method in real-world scenarios.

---

[¶]https://platform.openai.com/docs/models/gpt-3-5-turbo
[‖]https://platform.openai.com/docs/models/gpt-4-turbo-and-gpt-4
[**]https://docs.anthropic.com/en/docs/models-overview

Table 1: The detection performance (AUROC) of baselines and DPIC on three datasets generated by `ChatGPT`, `GPT-4`, and `Claude3`.

| Methods | ChatGPT | | | | GPT-4 | | | | Claude3 | | | |
|---|---|---|---|---|---|---|---|---|---|---|---|---|
| | XSum | Writing | PubMed | Avg. | XSum | Writing | PubMed | Avg. | XSum | Writing | PubMed | Avg. |
| RoBERTa-base | 0.9150 | 0.7084 | 0.6188 | 0.7474 | 0.6778 | 0.5068 | 0.5309 | 0.5718 | 0.8944 | 0.8036 | 0.3647 | 0.6876 |
| RoBERTa-large | 0.8507 | 0.5480 | 0.6731 | 0.6906 | 0.6879 | 0.3822 | 0.6067 | 0.5589 | 0.9027 | 0.7128 | 0.3579 | 0.6578 |
| RADAR | 0.9972 | 0.9593 | 0.7372 | 0.8979 | 0.9931 | 0.8593 | 0.8029 | 0.8851 | 0.9952 | 0.9438 | 0.8029 | 0.9139 |
| Likelihood | 0.9577 | 0.9739 | 0.8776 | 0.9364 | 0.7982 | 0.8553 | 0.8100 | 0.8212 | 0.9760 | 0.9744 | 0.9240 | 0.9581 |
| Entropy | 0.3305 | 0.1901 | 0.2766 | 0.2657 | 0.4364 | 0.3703 | 0.3296 | 0.3788 | 0.4109 | 0.0836 | 0.1686 | 0.2210 |
| LogRank | 0.9584 | 0.9656 | 0.8680 | 0.9307 | 0.7980 | 0.8289 | 0.7997 | 0.8089 | 0.9783 | 0.9732 | 0.9260 | 0.9592 |
| LRR | 0.9164 | 0.8962 | 0.7421 | 0.8516 | 0.7453 | 0.7040 | 0.6810 | 0.7101 | 0.9609 | 0.9598 | 0.8334 | 0.9180 |
| DNA-GPT(Neo-2.7) | 0.9040 | 0.9449 | 0.7598 | 0.8696 | 0.7267 | 0.8164 | 0.7163 | 0.7531 | 0.9071 | 0.9655 | 0.5911 | 0.8212 |
| DNA-GPT(ChatGPT) | 0.8396 | 0.7898 | 0.6722 | 0.7672 | 0.6146 | 0.6104 | 0.5745 | 0.5998 | 0.8560 | 0.8767 | 0.6729 | 0.8019 |
| DNA-GPT(Vicuna-7b) | 0.6992 | 0.6695 | 0.5639 | 0.6442 | 0.5594 | 0.5628 | 0.5366 | 0.5529 | 0.7241 | 0.7305 | 0.6001 | 0.6849 |
| NPR | 0.7845 | 0.9697 | 0.5483 | 0.7675 | 0.5211 | 0.8276 | 0.4976 | 0.6154 | 0.9232 | 0.9696 | 0.7746 | 0.8891 |
| DetectGPT | 0.4594 | 0.8008 | 0.3804 | 0.5469 | 0.3408 | 0.6542 | 0.3675 | 0.4542 | 0.4323 | 0.6800 | 0.7559 | 0.6227 |
| Fast-DetectGPT | 0.9907 | **0.9916** | 0.9021 | 0.9615 | 0.9064 | 0.9611 | 0.8498 | 0.9058 | 0.9942 | 0.9783 | 0.9035 | 0.9587 |
| DPIC(ChatGPT) | **1.0000** | 0.9821 | **0.9082** | **0.9634** | **0.9996** | **0.9768** | **0.9438** | **0.9734** | **1.0000** | **0.9950** | 0.9686 | **0.9878** |
| DPIC(Vicuna-7b) | 0.9976 | 0.9708 | 0.8990 | 0.9558 | 0.9986 | 0.9644 | 0.9394 | 0.9674 | 0.9992 | 0.9943 | **0.9690** | 0.9875 |

**Generalizability.** The detection performance of supervised methods, such as RADAR, exhibits significant performance differences in detecting different datasets. Taking ChatGPT-generated text as an example, the detection AUROC of RADAR on the Xsum dataset is 99.72%. However, the detection AUROC on the PubMed dataset is only 73.72%. This indicates that the detection performance of the supervised methods is susceptible to the domain dataset, which affects the generalizability. Zero-shot detection methods show more balanced detection performances on Xsum and PubMed datasets generated by ChatGPT. This suggests that the zero-shot method is more likely to maintain generalizability by extracting the intrinsic characteristics of LLMs. DPIC extracts the deep intrinsic characteristics of LLMs by decoupling the prompt, allowing the supervised detector to focus more on the intrinsic characteristics. This approach enables DPIC to achieve the desired generalization while maintaining high detection quality.

## 5.3 Ablation studies

We conducted ablation studies to reveal the impact of different similarity measurements and decoupling processes. Additionally, since the reconstructed prompts may differ from the original prompts, we discuss the impact of these differences on detection performance.

**Similarity measurement.** In this experiment, we explored the effects of different similarity measurements on the detection performance of the Siamese Network. Specifically, we experimented with absolute distance, dot product, and concatenation methods for combining feature vectors. The results are displayed in Table 2.

When combining the feature vectors by absolute distance and dot product, the detection AUROC scores are 89.81% and 93.84% in detecting the Writing dataset generated by ChatGPT. In contrast, when combining the feature vectors by concatenation, the AUROC score is 98.21%. These experimental results show that concatenation results in higher AUROC scores than other methods, which indicates that combining feature vectors by concatenation preserves more of the intrinsic characteristic information. In contrast, the other methods result in the loss of critical intrinsic details. This findings indicates that providing the detector with richer intrinsic characteristic information helps achieve higher detection quality.

**Prompt reconstruction and limitations** Since reconstructed prompts may differ from the original prompts, we discuss the impact of the prompt reconstruction process on detection performance in this part. We tested the semantic similarity of the candidate text and the regenerated text obtained by directly using the original prompt. We also test the semantic similarity of the candidate text and regenerated text obtained by the reconstructed prompt. We used `gte-Qwen1.5-7B-instruct`[††] for feature extraction, and then used cosine similarity to measure the degree of semantic similarity.

---

[††]https://huggingface.co/Alibaba-NLP/gte-Qwen1.5-7B-instruct

Table 2: Detection AUROC of different similarity measurement on Writing dataset generated by ChatGPT, GPT-4, and Claude3.

| Similarity Measurement | Decoupling by ChatGPT | | | Decoupling by Vicuna-7b | | |
|---|---|---|---|---|---|---|
| | ChatGPT | GPT-4 | Claude3 | ChatGPT | GPT-4 | Claude3 |
| Absolute Difference | 0.8981 | 0.8696 | 0.9206 | 0.7878 | 0.7476 | 0.8426 |
| Dot Product | 0.9384 | 0.894 | 0.9575 | 0.9299 | 0.9111 | 0.9753 |
| Concatenation (DPIC) | **0.9821** | **0.9768** | **0.9950** | **0.9708** | **0.9644** | **0.9943** |

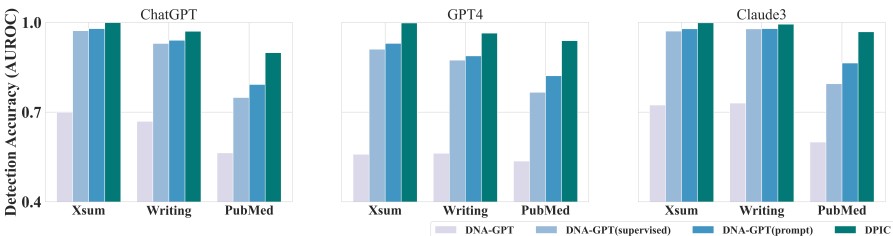

Figure 3: Detection AUROC of DNA-GPT, DNA-GPT (supervised), DNA-GPT (prompt), and DPIC. All regeneration processes are implemented using Vicuna-7b.

We use a part of HC3 test dataset generated by ChatGPT, GPT-4, claude3. The results are presented in Figure 4. From the result, we find that, compared to using the original prompt directly, the regenerated text obtained by the reconstructed prompt has a higher semantic similarity with the original text in texts generated by GPT-4 and Claude3.

We believe that this may be due to the following reasons: when users use LLM, the prompt is sometimes not detailed, which leads to the fact that the text generated by LLM may deviate from the original prompt. In other words, in the case where the prompt is not detailed, even if the same prompt is used, the text generated multiple times is prone to larger deviations. However, when LLM is used to regenerate the prompt from a candidate text, the reconstructed prompt is reconstruct from the candidate text, which makes the prompt more semantically consistent with the candidate text. Thereby, when we use the reconstructed prompt for text regeneration, the regenerated text can show more semantic similarity with the candidate text, which is more helpful in reducing the impact of semantics on the detection process.

However, directly using LLMs for regenerating for semantic disentanglement can indeed introduce some time delays and costs, which we will discuss in the Limitations Section 6.

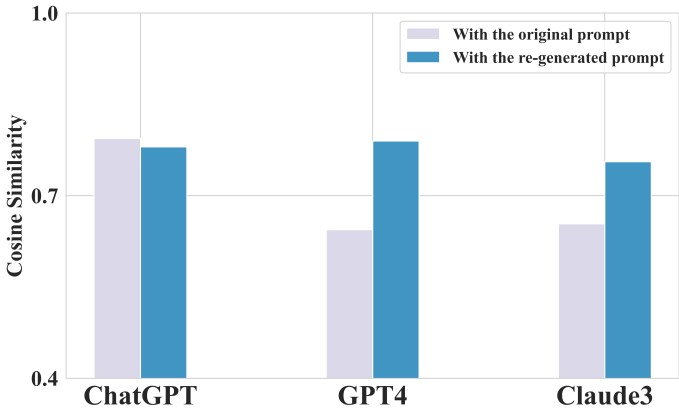

Figure 4: Cosine Similarity of the candidate text and the regenerated text obtained by different prompts.

**Decoupling Process**    In the decoupling process, given a candidate text, we initially reconstruct the prompt based on the candidate text. The reconstructed prompt is then used for the auxiliary LLM to obtain the regenerated text. Then, we use the embeddings of the candidate text and the regenerated text as the discriminative feature to train a classifier. DNA-GPT [42] shares a similar idea. Specifically, given a candidate text, DNA-GPT truncates from the half position of the text length and then completes the text based on the truncated text using an auxiliary LLM. Unlike us, DNA-GPT calculates the divergence between the candidate text and the completion of the truncated text and uses a zero-shot classifier for detection. Another significant difference is that there is no semantic prompt guidance in the completion process. In order to fully illustrate the effectiveness of our method, we explore the impact of supervised learning and prompt guidance on detection performance using DPIC and DNA-GPT as examples.

First, we compare the impact of zero-shot and supervised methods on the detection performance using DNA-GPT. Specifically, given a candidate text, DNA-GPT truncates from the half position of the text length and then completes the truncated text using an auxiliary LLM. Then, we use the embeddings of the candidate text and the completion of the truncated text as the discriminative feature to train a Siamese Network to classifier; the detailed experimental setup is consistent with the description in Section 5.1. The results are displayed in Figure 3, comparing DNA-GPT and DNA-GPT (supervised). The average detection AUROC of DNA-GPT (supervised) is 88.39%, 85.50%, and 91.51% on datasets generated by ChatGPT, GPT-4, and Claude3, respectively. In contrast, the average detection AUROC of DNA-GPT is 64.42%, 55.29%, and 68.49%. DNA-GPT (supervised) shows a substantial improvement compared to DNA-GPT, demonstrating that supervised methods help achieve higher detection AUROC. However, DNA-GPT (supervised) exhibits a significant decrease in detection AUROC on the PubMed dataset compared to the Xsum and Writing datasets, indicating that generalizability is affected when using supervised methods.

Afterward, we evaluated the impact of the prompt semantic guidance on the detection performance and generalizability of DNA-GPT (supervised). Specifically, given a candidate text, we reconstruct the prompt based on the candidate text and complete the truncated text based on the prompt using an auxiliary LLM. Then, we measure the difference between the candidate text and the prompt-based completion of the truncated text and use a Siamese Network as the classifier. The results are displayed in Figure 3, comparing DNA-GPT (supervised) and DNA-GPT (prompt). Compared to DNA-GPT (supervised), DNA-GPT (prompt) achieves a more balanced detection performance across the three datasets. On the PubMed dataset, DNA-GPT (prompt) achieves detection accuracies of 79.28%, 82.18%, and 86.46%, while DNA-GPT (supervised) achieves only 74.93%, 76.66%, and 79.52%, respectively. It demonstrates the importance of prompts in regeneration, as they help reduce semantic bias between the regenerated text and the candidate text, thereby mitigating the impact of the dataset domain on the generalizability of the detector.

Finally, we compare the detection performance with the truncated-completion method and the regeneration method on the detection performance, and the results are displayed in DNA-GPT (prompt) and DPIC in Figure 3. As shown, the regeneration method of DPIC achieves better detection quality and better generalization.

## 5.4   Robustness

Existing research [19, 33] has pointed out that the previous methods experience a reduction in performance in complex scenarios where the text to be detected is subjected to interference. To better understand how DPIC performs in real-world scenarios, we evaluate our detection method under two modification methods.

The first one is the proposed paraphrasing attack called DIPPER [19] (or Discourse Paraphrase). DIPPER is an 11B-parameter paraphrase generation model built by fine-tuning T5-XXL. It can paraphrase paragraph-length texts, re-order content, and optionally leverage context such as input prompts. The second interference we consider is a method that is more accessible to the broader audience of large language models and does not require specialized knowledge, namely the back-translation attack. Back-translation refers to the action of *translating a work that has previously been translated into the same language*. We employed DeepL Translator [‡‡] to translate the given English text into Chinese, followed by a subsequent translation back into English.

---

[‡‡] https://www.deepl.com/en/docs-api/

Table 3: Detection performance of DPIC and the baselines in detecting Xsum dataset generated by ChatGPT, GPT-4, and Claude3 with interference.

| Methods | ChatGPT | | | GPT-4 | | | Claude3 | | |
|---|---|---|---|---|---|---|---|---|---|
| | Ori. | DIPPER | Back-translate | Ori. | DIPPER | Back-translate | Ori. | DIPPER | Back-translate |
| RoBERTa-base | 0.9150 | 0.8148 | 0.8379 | 0.6778 | 0.6469 | 0.7536 | 0.8944 | 0.8120 | 0.8052 |
| RoBERTa-large | 0.8507 | 0.7884 | 0.6853 | 0.6879 | 0.6833 | 0.6660 | 0.9027 | 0.8153 | 0.7583 |
| RADAR | 0.9972 | 0.9964 | 0.9801 | 0.9931 | 0.9924 | 0.9608 | 0.9952 | 0.9940 | 0.9701 |
| Likelihood | 0.9577 | 0.8438 | 0.9306 | 0.7982 | 0.6296 | 0.8449 | 0.9760 | 0.9080 | 0.9446 |
| Entropy | 0.3305 | 0.4514 | 0.3008 | 0.4364 | 0.5552 | 0.3705 | 0.4109 | 0.4978 | 0.3639 |
| LogRank | 0.9584 | 0.8596 | 0.9260 | 0.7980 | 0.6432 | 0.8436 | 0.9783 | 0.9256 | 0.9488 |
| LRR | 0.9164 | 0.8448 | 0.8621 | 0.7453 | 0.6607 | 0.8003 | 0.9609 | 0.9240 | 0.9243 |
| DNA-GPT(Neo-2.7) | 0.9040 | 0.7733 | 0.8624 | 0.7267 | 0.5595 | 0.7776 | 0.9071 | 0.7876 | 0.8399 |
| DNA-GPT(ChatGPT) | 0.8396 | 0.7910 | 0.7975 | 0.6146 | 0.5454 | 0.6070 | 0.8560 | 0.7996 | 0.7814 |
| DNA-GPT(Vicuna-7b) | 0.6992 | 0.6528 | 0.6175 | 0.5594 | 0.5523 | 0.5827 | 0.7241 | 0.6403 | 0.6321 |
| NPR | 0.7845 | 0.5648 | 0.8050 | 0.5211 | 0.3006 | 0.6820 | 0.9232 | 0.7860 | 0.9042 |
| DetectGPT | 0.4594 | 0.3074 | 0.5417 | 0.3408 | 0.1823 | 0.4530 | 0.4323 | 0.3283 | 0.5273 |
| Fast-DetectGPT | 0.9907 | 0.9536 | 0.9711 | 0.9064 | 0.8057 | 0.9137 | 0.9942 | 0.9720 | 0.9860 |
| DPIC(ChatGPT) | **1.0000** | **1.0000** | **0.9972** | **0.9996** | **0.9991** | **0.9931** | **1.0000** | 0.9996 | **0.9979** |
| DPIC(Vicuna-7b) | 0.9976 | 0.9980 | 0.9889 | 0.9986 | 0.9969 | 0.9903 | 0.9992 | **0.9996** | 0.9966 |

We present the detection performance of DPIC and baselines in detecting Xsum dataset generated by ChatGPT, GPT-4, and Claude3 with interference in Table 3. RADAR shows the smallest decrease among baselines against DIPPER attacks, especially for text generated by GPT-4, with a decrease of 00.07%, illustrating the robustness of RADAR in incorporating adversarial networks into detection. However, our method still maintains optimal detection performance after both DIPPER and back-translation attacks. The detection AUROC of DPIC is 100% and 99.72% for detecting the Xsum dataset generated by ChatGPT under DIPPER and back-translation attacks, respectively, indicating that our method is more robust in real-world scenarios. DIPPER and back-translation attacks mainly alter the text semantically. Our method decouples the prompt-guided semantic information and the intrinsic characteristics of the generative model, rendering semantic attacks ineffective and ensuring robustness.

## 6  Limitations

We mitigated the risk of benchmark contamination in the testing phase by ensuring different sources for the training and test sets. However, recent studies [7, 40, 34] have indicated that benchmark contamination may be widespread in evaluating large models, which is a limitation not addressed in this paper.

Another limitation of our method is the time required in the decoupling process. The regeneration process significantly extends the processing time for each text segment. We selected advanced training-based and zero-shot detection methods and tested memory and time consumption. Our method does require more memory and time, compared to those that only input candidate text for classification, primarily due to the regeneration component involving the LLM. Detailed information regarding this can be found in the Appendix A.

## 7  Conclusion

We reconsider the detection methods for LLM-generated text and propose a decoupling-based detection method, DPIC. This method extracts deep intrinsic characteristics of the black-box model generated texts. Through the decoupling process, we provide the detector with a reconstructed prompt covering the topic of the candidate text, allowing the detector to focus on deep intrinsic characteristics, thereby achieving better generalization performance while maintaining detection quality. Experimental results further demonstrate that DPIC significantly improves detection performance by 6.76% and 2.91% when detecting texts generated by GPT-4 and Claude 3, respectively.

# 8 Acknowledgements

This work was supported in part by the National Natural Science Foundation of China under Grant U2336206, Grant 62472398, Grant 62102386, Grant 62121002, Grant U20B2047 and by Open Foundation of Key Laboratory of Cyberspace Security, Ministry of Education (No.KLCS20240207).

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
