# OpenReview forum: "DPIC: Decoupling Prompt and Intrinsic Characteristics for LLM Generated Text Detection"
_NeurIPS.cc/2024/Conference — NeurIPS 2024 poster_

### Official Review · Reviewer_eWrN · 2024-06-19

**Soundness:** 3
**Presentation:** 3
**Contribution:** 3
**Rating:** 7
**Confidence:** 4

**Summary:**

In this paper, the authors present DPIC, a novel model for detecting LLM-generated text, which centers on having an auxiliary LLM reverse-generate prompt on candidate text, and then letting the LLM re-generate the answers to the prompt and classify them based on the similarity between the candidate and the re-generated text. The authors experimented it with competitors on multiple datasets and claimed that they achieved better performance.

**Strengths:**

- This paper is well written and detailed.

- The experiments for the model were adequate.

- The core idea of the model is easy to understand and interesting.

**Weaknesses:**

- Lack of detail in evaluation metrics.

- The testing model is underrepresented.

- Details of the model's consumption of computational resources are missing.

**Questions:**

1. In the manuscript, I did not find out any details about the AUROC evaluation metrics used, such as its formula, or the evaluation steps. I understand that this is a commonly used evaluation metric, but an academic paper should be complete and detailed. The authors only briefly indicated its mathematical meaning, which I think is not sufficient.

2. For the testing part of the models, the authors used three representative commercial closed-source models: gpt-3.5, gpt-4, and claude3 to generate text for specific prompts for model testing. While it can be argued that closed-source models are more difficult to access than open-source models to infer that it is more challenging to recognize text generated by closed-source models, this inference is from the perspective of detection model builder, and applied to a real-world scenario, even if the candidate text is open-source LLM-generated, will the detection model have easy access to the internal parameters used to generate this text? I don't think the authors should make too much of a distinction between closed-source and commercial models for the task of detecting LLM-generated text. So I suggest the authors to add some open-source models to the test models, which will make the conclusions more convincing, and I think the readers are also interested in the differences between texts generated by commercial closed-source models and open-source models. I understand the infeasibility of conducting new experiments in a short period of time, so I would suggest that the authors mention this in the limitations section.

3. The authors mention in question 8 of the checklist that they provide enough details about computational resource consumption in Section 5.1, but I didn't find them in the corresponding section. I think that for this kind of detection task, comparing the running efficiency of the proposed model and the competitors is also an important part. In particular, I'd be interested to know the resource consumption comparison between DPIC models and those that only input candidate text for classification.

**Limitations:**

I think the statement made by the author in the limitations section (Appendix A) is not sufficient. It seems that the author is only analyzing the results of the experiment in the prompt reconstruction part by mentioning the possible scenarios that may occur when using the model. I would suggest that the authors flesh out this section from the perspective of the model's deployment in a real-world environment and its impact on society. Additionally, I suggest that the authors include an analysis with some of the current techniques that help AI-generated text escape detection, to test whether the model will be defeated by these models.

---

> ### Author Rebuttal · Authors · 2024-08-07
>
> We thank the reviewer for the time and expertise you have invested in these reviews. We are delighted to receive positive feedback that our work provides a solid contribution to the field. Below we provide point-by-point responses to your comments and questions.
>
> ---
>
> - **Question 1**: Lack of detail in evaluation metrics, ROC definition.
>
>     **Response:** The receiver operating characteristic curve, or ROC curve, is the plot of the true positive rate (TPR) against the false positive rate (FPR) at each threshold setting. For a detector $f$, its AUROC can be expressed by the following equation:
>
>     $AUC(f) = \frac{\sum_{t_0\in{\mathcal{D}^{0}}}\sum_{t_1\in{\mathcal{D}^{1}}}\mathbf{1}\left[f(t_0)<f(t_1)\right]}{\vert\mathcal{D}^{0}\vert\cdot\vert\mathcal{D}^{1}\vert}$
>
>     where $\mathbf{1}\left[f(t_0)<f(t_1)\right]$ denotes an *indicator function* which returns 1 if $f(t_0)<f(t_1)$ otherwise return 0; $\mathcal{D}^{0}$ is the set of negative examples, and $\mathcal{D}^{1}$ is the set of positive examples.
>
>     We will add this part to the final version of the paper.
>
> ---
>
> - **Question 2**: I suggest the authors add some open-source models to the test models, which will make the conclusions more convincing.
>
>     **Response:** Thanks for your suggestion! I agree with your viewpoint that detecting open-source LLM generated texts is meaningful. We carried out detection on two advanced open-source LLMs, Qwen1.5-7B-Chat [8] and Llama-3.1-405B-Instruct [9]. We also included comparisons with RADAR and Fast-DetectGPT, which are advanced training-based and zero-shot methods. We achieved significant detection performance, demonstrating the practicality of our method. We will include this part in the final version.
>
>     |  | Qwen-1.5 |  | | |  | Llama-3.1|  |  |  |
>     | --- | --- | --- | --- | --- | --- | --- | --- | --- |--- |
>     |  | Xsum | Writing | Pub | Avg. | | Xsum | Writing | Pub | Avg. |
>     | Fast-DetectGPT | 0.9981 | **1.0000** | 0.7978 | 0.9319 | | 0.9986 | **0.9908** | 0.9251 | 0.9715 |
>     | RADAR | 0.9951 | 0.8752 | 0.5476 | 0.8059 | | 0.9930 | 0.8908 | 0.5183 | 0.8007 |
>     | DPIC | **0.9999** | 0.9865 | **0.9967** | **0.9943** | | **0.9988** | 0.9809 | **0.9526** | **0.9774** |
>
> ---
>
> - **Question 3**: Running efficiency and Resource Consumption comparison between DPIC and detector.
>
>     **Response:** Thank you for your suggestion. We selected advanced training-based and zero-shot detection methods and tested memory and time consumption, as shown in the table below. Our method does require more memory and time, compared to those that only input candidate text for classification, primarily due to the regeneration component involving the LLM.
>
>     |  | Time Cost | Memory Consumption |
>     | --- | --- | --- |
>     | Fast-DetectGPT | 0.273s | 25447 MB |
>     | RADAR | 0.120s | 1355 MB |
>     | Ghostbuster | 2.493s | <500MB |
>     | DPIC | 2.665s | 28105 MB |
>
>     We have also implemented strategies to minimize costs and enhance real-world applicability, such as substituting ChatGPT with Vicuna-7b, which allows for the entire process to be locally deployed at the cost of a slight decrease (0.46%) in the average detection AUROC. Besides, we will address cost considerations as part of the limitations.
>
>     Despite the associated cost, our method holds great practical value and significance. Generally, for many detection tasks that involve large amounts of generated text, a multi-stage detection strategy is practically effective. Initially, a fast detection model with lower accuracy is used for preliminary screening. Once potentially generated text is identified, a more accurate detection model is employed in the second stage for detailed verification. Our method is well-suited as an accurate detection model for this second stage. Additionally, our method offers direct practical value in situations where time constraints are relaxed and the volume of candidate texts is manageable, such as in academic plagiarism detection.
>
> ---
>
> - **Limitations 1.1:** I would suggest that the authors flesh out this section from the perspective of the model's deployment in a real-world environment and its impact on society.
>
>     **Response:** The ability to accurately detect LLM-generated text is critical for realizing the full potential of natural language generation (NLG) while minimizing serious consequences such as phishing, disinformation, and academic dishonesty. From the perspective of the end users, LLM-generated text detection could increase trust in NLG systems and encourage adoption. For machine learning system developers and researchers, the detector can aid in tracing generated text and preventing unauthorized use. Given its significance, our DPIC method can deepen the academic and industrial understanding of the mechanisms behind LLM-generated text detection. When deployed in a real-world environment, DPIC, like all other detection methods, requires GPU computing power. Balancing resource consumption and performance gains is also a problem worth investigating. We will flesh out limitation part from the perspective of the model's deployment in a real-world environment and its impact on society. Thank you again for your valuable suggestion!
>
> - **Limitations 1.2:** An analysis with some of the current techniques that help AI-generated text escape detection, to test whether the model will be defeated by these models.
>
>     **Response:** For robustness experiments against evasion, we have also added robustness tests against OUTFOX [5]. The results in **Table 1** of the attached pdf show that DPIC demonstrates superior performance, which verifies its effectiveness. Besides, we will add discussion of AI-generated text escape detection in the limitation part and point out the urgency of robust detector research.
>
> ### **References**
> Please see **Rebuttal References** in the global response.

---

> ### Comment · Reviewer_eWrN · 2024-08-10
>
> Thank you for the clarification, I am satisfied with it. In addition I would like the authors to include, in the limitations or conclusion section, an assessment of the risk of benchmark contamination in the evaluation process, which would make the analysis of the experimental results more robust, as can be seen in the following works.
>
> [1] Unveiling the Spectrum of Data Contamination in Language Models: A Survey from Detection to Remediation, ACL 2024.
>
> [2] Benchmark Data Contamination of Large Language Models: A Survey, arXiv 2024.
>
> [3] NLP Evaluation in trouble: On the Need to Measure LLM Data Contamination for each Benchmark, EMNLP Findings 2023.

---

> > ### Author Response · Authors · 2024-08-12
> > **Thank you for your response**
> >
> > We are delighted to learn that the reviewer is satisfied with our response! We have read the works you mentioned, and it is crucial to consider whether the benchmark might be contaminated during the evaluation process. We will add the assessment of the risk of benchmark contamination into the limitations and conclusion sections to strengthen our manuscript, as the reviewer suggested. Thank you again for the time and expertise you have invested in these reviews.

---

### Official Review · Reviewer_Av5h · 2024-07-11

**Soundness:** 3
**Presentation:** 3
**Contribution:** 3
**Rating:** 7
**Confidence:** 4

**Summary:**

This paper addresses the problem of detecting texts generated by large language models (LLMs), which is a crucial issue considering the potential misuse of such models. The authors propose a novel method, DPIC (Decoupling Prompt and Intrinsic Characteristics), which aims to extract the intrinsic characteristics of black-box models, as traditional detection methods requiring access to the model's interior are not feasible. DPIC uses an auxiliary LLM to reconstruct the prompt of a candidate text and regenerate a text from it, allowing the detector to focus on the intrinsic characteristics of the generative model. The similarity between the candidate and regenerated texts is then used as a detection feature. Results show that DPIC outperforms baseline methods, achieving an average improvement of 6.76% and 2.91% in detecting texts from different domains generated by GPT4 and Claude3, respectively.

**Strengths:**

The paper presents a novel approach to distinguishing between machine-generated and human-written texts by decoupling intrinsic characteristics from prompts. This innovative method offers new insights into the essential differences between these two types of text, marking a significant contribution to the field. One of the major strengths of this approach is its applicability to proprietary models. Given the usual inaccessibility to model parameters, working with such black-box models is challenging. The paper also demonstrates robustness across various datasets, source models, and paraphrasing attacks, suggesting that the proposed method is capable of maintaining performance under different conditions and adversarial scenarios. Another significant strength is the method's performance in terms of AUROC. It achieves significantly higher AUROC scores compared to previous zero-shot methods, although the paper acknowledges that comparing a method requiring training to zero-shot methods may not be entirely fair.

**Weaknesses:**

There are a few areas that could be improved. Firstly, the comparison with zero-shot methods seems potentially unfair since the proposed method requires training. Including trained detectors as baselines, for instance, using the same LLM used by DPIC to train a classifier directly, might provide a more balanced comparison. Secondly, the paper lacks a detailed analysis of the method's robustness on different lengths of text and languages. This could limit the applicability of the method in real-world scenarios. Lastly, the content of Section 3 appears misplaced as it discusses experimental settings rather than threat models. A reorganization or further clarification in this section would enhance the overall coherence of the paper.

**Questions:**

Figure2, can we use a pre-trained Siamese Network to measure the similarity in DPIC? Then, we do not need a training here.

LN121, the section 3 looks strange because it does not provide a threat model but some experimental settings.

Table1, I would expect to see more comparsion with trained detectors given that DPIC is also a trained detector.

Figure3, it seems the major contribution is from the supervised training, while the contribution from the prompt is much smaller. Why?

Figure3, the improvement of DNA-GPT(prompt) compared to DNA-GPT(supervised) mainly happens on PubMed. Is it caused by the QA style of the dataset?

**Limitations:**

Yes.

---

> ### Author Rebuttal · Authors · 2024-08-07
>
> We thank the reviewer for the time and expertise you have invested in these reviews. We are delighted to receive positive feedback that our work provides a significant contribution to the field, especially since our innovative method offers new insights into the essential differences between human and AI-generated texts, and the superior generalization detection performance of our method for black-box models. This is indeed our original aspiration.
>
> Below we provide point-by-point responses to your comments and questions.
>
> ---
>
> - **Weaknesses 1 & Question 3**: Given that DPIC is also a trained detector, I would expect to see more comparisons with trained detectors. For instance, using the same LLM used by DPIC to train a classifier directly.
>
>     **Response:** This is a great suggestion. Following your advice, we added a gte-Qwen1.5-7B-instruct[6] classifier (the same LLM used by DPIC) and trained it from scratch using the same training datasets as DPIC. We also added comparisons with other training-based methods, including Ghostbuster[1], Fingerprints[2], and CoCo[4].  These results are shown in the below table and **Table 1** in the attached PDF. It can be seen that our method still achieves the best detection performance compared to other baseline methods, especially in terms of detection generalization, which is a shortcoming of training-based detectors.
>
>     |  | ChatGPT |  |  |  | GPT4 |  |  |  | Claude3 |  |  |  |
>     | --- | --- | --- | --- | --- | --- | --- | --- | --- | --- | --- | --- | --- |
>     |  | XSum | Writing | PubMed | Avg. | XSum | Writing | PubMed | Avg. | XSum | Writing | PubMed | Avg. |
>     | gte-Qwen1.5-7B-instruct Classifier | 0.9964 | 0.9505 | 0.7827 | 0.9098 | **0.9999** | 0.9460 | 0.7996 | 0.9151 | 0.9998 | 0.9879 | 0.9338 | 0.9738 |
>     | DPIC | **1.0000** | **0.9821** | **0.9082** | **0.9634** | 0.9996 | **0.9768** | **0.9438** | **0.9734** | **1.0000** | **0.9950** | **0.9686** | **0.9878** |
>
>      ---
>
>
> - **Weaknesses 2:** The paper lacks a detailed analysis of the method's robustness to different text lengths and languages.
>
>     **Response:** This is a great suggestion. Following your advice, we evaluated DPIC's robustness to the number of words in text and different languages. First, we divided the texts in the Xsum, Writing, and PubMed datasets based on word count and tested DPIC's average detection accuracy. From the results shown in **Figure 2** in the attached PDF, DPIC performs best when the word count is larger than 100.
>
>     We selected Chinese and Urdu according to the M4 benchmark[7] for **different languages**, achieving detection AUROCs of 0.9895 and 0.9830, respectively. This is thanks to the regenerative and encoder models we used, which support multiple languages. These results demonstrate the practicality of DPIC in real-world scenarios, and we will add this part in the final version.
>
> ---
>
> - **Weaknesses 3 & Question 2:** The content of Section 3 appears misplaced as it discusses experimental settings rather than threat models.
>
>     **Response:** Sorry for the confusion. We will restate the threat model in terms of the attacker's capabilities and objectives, as well as the defender's capabilities and objectives in the final version. The attacker’s objective is to use large language models to generate text that impersonates human-written text,  and his capability is using the existing large language model, e.g., GPT4 and Claude3.  The defender’s objective is to distinguish between human and generated text. In this paper, we focus on the black-box scenario, and the capability of the defender is he can only use the text input and output of the potential source LLMs.
>
> ---
>
> - **Question 1:** In Figure 2, Can we use a pre-trained Siamese Network to measure the similarity in DPIC?
>
>     **Response:** This is a good question. Following your advice, we measure the cosine similarity of the embeddings of the pre-trained Siamese Network, and evaluate the detection AUROC.  As shown in the results below, we take the datasets generated by Claude3 as an example. From the results, it can be seen directly using cosine similarity cannot yield considerable effective detection.
>
>     |  | XSum | Writing | PubMed | Avg. |
>     | --- | --- | --- | --- | --- |
>     | No training | 0.4801 | 0.6913 | 0.6931 | 0.6215 |
>     | DPIC | **1.0000** | **0.9950** | **0.9686** | **0.9878** |
>
> ---
>
> - **Question 4 & Question 5:** Figure3, it seems the major contribution is from the supervised training, while the contribution from the prompt is much smaller. Why? Figure3, the improvement of DNA-GPT(prompt) compared to DNA-GPT(supervised) mainly happens on PubMed. Is it caused by the QA style of the dataset?
>
>     **Response:**
>     We will answer these questions together. In Figure 3, DNA-GPT(supervised) utilizes the candidate text and its truncated and regenerated text to train a Siamese network classifier, which, compared to the original DNA-GPT's N-gram classification, can better decouple the intrinsic features of the text from its semantic information, achieving a significant performance improvement.
>
>     The contribution from the prompts is less significant in the Xsum and Writing datasets, but more substantial in QA style dataset PubMed. The reason is indeed as you thought, the regenerated prompts from DPIC can assist DNA-GPT(prompt) in obtaining regenerated text that aligns more closely with the PubMed’s QA style, thereby, the improvement is more pronounced on this dataset.
>
>     The above analysis indicates the effectiveness of both components of our DPIC method: the supervised training based on a siamese network and the regenerated prompt procedure. I hope this explanation resolves your concerns. We also appreciate the time you have taken to thoroughly understand our method.
>
> ---
>
> ### **References**
> Please see **Rebuttal References** in the global response.

---

> > ### Comment · Reviewer_Av5h · 2024-08-12
> >
> > Thanks for the clarification, which addresses my major concerns.

---

> > > ### Author Response · Authors · 2024-08-12
> > > **Thank you for your response**
> > >
> > > We are delighted that our response addresses the reviewer's major concerns. Again, we thank the reviewer for your valuable and positive feedback and comments.

---

### Official Review · Reviewer_XAPE · 2024-07-12

**Soundness:** 3
**Presentation:** 3
**Contribution:** 3
**Rating:** 6
**Confidence:** 4

**Summary:**

This paper proposes a novel method named DPIC (Decoupling Prompt and Intrinsic Characteristics) for detecting texts generated by LLMs. The authors posit that generated texts are a coupled product of prompts and intrinsic characteristics, and suggest that decoupling these two elements can enhance detection quality and generalization ability. Specifically, the DPIC method employs an auxiliary LLM to reconstruct the prompt of the candidate text, and then uses this prompt to regenerate the text. By comparing the similarity between the candidate text and the regenerated text, the detector can better focus on the intrinsic characteristics of the generating model. Experimental results show that, compared to baseline methods, DPIC improves the detection of texts generated by GPT-4 and Claude-3 in different domains by an average of 6.76% and 2.91%, respectively.

**Strengths:**

- A new perspective is proposed, improving detection performance by decoupling prompts and intrinsic characteristics, with strong motivation and description. More importantly, this method can handle black-box detection scenarios.
- Experimental results demonstrate that the DPIC method significantly enhances detection quality and generalization ability, especially when dealing with black-box models.

**Weaknesses:**

- DPIC involves multiple steps, including prompt reconstruction and text regeneration, which significantly increase computational costs and resource overhead in practical applications. An ablation analysis on the impact of the training sample size would help readers further understand and evaluate the DPIC method.
- DPIC is essentially a supervised method and should be compared fairly with methods based on RoBERTa (not the OpenAI detector trained on GPT-2 but a RoBERTa classifier trained from scratch).
- Supervised methods perform well within their domain but have poor out-of-distribution generalization ability, which lacks discussion in the paper. For example, the performance of DPIC trained on XSum on Writing, and the performance of DPIC trained on ChatGPT re-generated data on Claude-3. This is crucial for highlighting the effectiveness of DPIC in practical applications.

**Questions:**

How is the accuracy of prompt reconstruction evaluated? Did the authors verify the consistency between the reconstructed prompts and the original prompts? If there is a significant difference between the reconstructed prompts and the original prompts, how much impact would that have on the detection results?

**Limitations:**

The authors did not discuss the out-of-distribution generalization ability of DPIC.

---

> ### Author Rebuttal · Authors · 2024-08-07
>
> We sincerely thank you for your insightful reviews of our manuscript. Below, we provide point-by-point responses to your comments and questions.
>
> - **Limitations & Weaknesses 3:** The authors did not discuss the out-of-distribution generalization ability of DPIC.
>
>     **Response:** Actually, we have already considered the out-of-distribution generalization in our original paper, including the generalization to three datasets and different LLMs. Other reviewers have also praised the generalizability of DPIC. As we described in lines 198-201 and 208-215 of our paper: for training, we used only HC3 dataset; For testing, we used three datasets that are not in the domains covered in the HC3 dataset to evaluate the generalizability of detectors. We also use different LLMs, e.g., Claude3, to generate texts on the test datasets. I hope this response can clear up your misunderstandings.
>
>     ---
>
> - **Weaknesses 2**: DPIC is essentially a supervised method and should be compared fairly with methods based on RoBERTa classifier trained from scratch.
>
>     **Response**: This is a great suggestion. Following your advice, we trained RoBERTa-base and RoBERTa-large classifier **from scratch using the same training datasets as DPIC**. We also added comparisons with other training-based methods, including Ghostbuster[1], Fingerprints[2], and CoCo[4]. These results are shown in the below table and **Table 1 in the attached PDF**. DPIC still achieves the best detection performance compared to other methods, further demonstrating the effectiveness of DPIC. We will include this in the final version of the paper.
>
>     |  | ChatGPT |  |  |  | GPT4 |  |  |  | Claude3 |  |  |  |
>     | --- | --- | --- | --- | --- | --- | --- | --- | --- | --- | --- | --- | --- |
>     |  | XSum | Writing | PubMed | Avg. | XSum | Writing | PubMed | Avg. | XSum | Writing | PubMed | Avg. |
>     | Roberta-base-trained from scratch | 0.6828 | 0.8298 | 0.7582 | 0.7569 | 0.6830 | 0.8235 | 0.7113 | 0.7392 | 0.9699 | 0.9897 | 0.9379 | 0.9658 |
>     | Roberta-large-trained from scratch | 0.9066 | 0.9327 | 0.7450 | 0.8614 | 0.7360 | 0.8180 | 0.7448 | 0.7662 | 0.9678 | 0.9798 | 0.8670 | 0.9382 |
>     | DPIC | **1.0000** | **0.9821** | **0.9082** | **0.9634** | **0.9996** | **0.9768** | **0.9438** | **0.9734** | **1.0000** | **0.9950** | **0.9686** | **0.9878** |
>
>     ---
>
> - **Weaknesses 1**: DPIC involves multiple steps, including prompt reconstruction and text regeneration, which significantly increase computational costs and resource overhead in practical applications. An ablation analysis on the impact of the training sample size would help readers further understand and evaluate the DPIC method.
>
>     **Response**: Our method’s feature extraction process incurs certain costs, but the benefits it offers are significant. As Table 1 in our original paper shows, our method demonstrates superior generalization compared to existing approaches, achieving an average detection AUROC of 0.9806 in identifying texts from diverse domains generated by two commercial closed-source models: GPT4 and Claude3. The superior performance brings practical value. Our method can be deployed in the second stage of a funnel-shaped multi-stage detection process, and can also offers direct practical value in situations where time constraints are relaxed and the volume of candidate texts is manageable, such as in academic plagiarism detection.
>
>     Following your advice, we tested the impact of the training sample size (500-2500) on DPIC’s detection performance. The results in **Figure 1 in the attached PDF** indicate that our method achieves relatively excellent detection performance even with a smaller sample size because DPIC captures intrinsic features and reduces the impact of training domains.
>
> ---
>
> - **Question**: How is the accuracy of prompt reconstruction evaluated? Did the authors verify the consistency between the reconstructed prompts and the original prompts? If there is a significant difference between the reconstructed prompts and the original prompts, how much impact would that have on the detection results?
>
>     **Response:**
>     In our original paper, we did not directly measure the consistency between the reconstructed prompts and the original prompts.
>     Since DPIC aims to mitigate the influence of semantics in the detection process, we directly measured the semantic consistency between the original text and the regenerated text generated by the reconstructed prompts and the original prompts, to evaluate the effectiveness of the reconstructed prompts.
>     The results in Figure 4 of the original paper show that the regenerated text obtained by the reconstructed prompt has a higher semantic similarity with the original text in texts generated by GPT4 and Claude3. The underlying reason can be revealed from Table 4 that our question generation prompt makes the reconstructed prompts not only cover the original prompt but also have more details, helping LLM generate more similar text with respect to the original text. In other words, there is no significant difference between the reconstructed prompts and the original prompts.
>
>     Following your advice, we have evaluated the detection results of different prompts on PubMed dataset with Claude3 model, including the original prompts and our reconstructed prompts, achieving a detection AUROC of 0.9543 and 0.9686, respectively. The results show that DPIC performs better than using the original prompt, indicating that the prompt will impact the detection performance.
>
>     We will clarify this in the final version and thanks for your valuable suggestions.
>
> ---
>
> ### **References**
>
> Please see **Rebuttal References** in the global response.

---

> ### Comment · Reviewer_XAPE · 2024-08-09
> **Thanks for the clarification**
>
> Thanks for the clarification, which addresses some of my key concerns. I have revised my score accordingly.
>
> Honestly, the motivation for this paper is excellent. However, my main concern is whether the regenerated text consistently maintains a high semantic similarity with the original. I'm not entirely convinced yet. Providing more support and analysis on this point could strengthen the manuscript.

---

> > ### Author Response · Authors · 2024-08-12
> > **Official Comment by Authors**
> >
> > We sincerely appreciate your recognition of our paper's motivation. Below, I will answer your question on 'whether the regenerated text consistently maintains a high semantic similarity with the original.'
> >
> > We used the gte-Qwen1.5-7B-instruct model to extract embeddings from the original text and the regenerated text, and then measured their semantic similarity using cosine similarity. We evaluated 9 datasets and obtained the values of cosine similarity for Average, Standard Deviation, First Quartile, Second Quartile, and Third Quartile.
> >
> > To better understand the degree of semantic similarity reflected by the value of cosine similarity, we introduce the baseline similarity for reference, which is defined as the cosine similarity between embeddings of human and AI-generated texts for the same prompt. Since the texts correspond to the same prompt, their semantic similarity can be regarded as at a high level.
> >
> > As shown in the table below, the average similarity exceeds the baseline similarity across all 9 datasets, which indicates that the original and regenerated texts achieve high semantic similarity. Additionally, the standard deviation values range from 0.11 to 0.15, which is relatively small compared to the average semantic similarity. This indicates that most of the texts maintain a high level of semantic similarity with the regenerated text.
> >
> > I hope this explanation addresses your concern. If you have any further concerns or questions about our work, we are happy to discuss them with you. We will also add this part to strengthen the manuscript. Thank you again for the time and expertise you have invested in these reviews.
> >
> > |  | ChatGPT |  |  | GPT4 |  |  | Claude3 |  |  |
> > | --- | --- | --- | --- | --- | --- | --- | --- | --- | --- |
> > |  | XSum | Writing | PubMed | XSum | Writing | PubMed | XSum | Writing | PubMed |
> > | Baseline Similarity | 0.5042 | 0.3739 | 0.5207 | 0.5065 | 0.3760 | 0.5225 | 0.2816 | 0.2538 | 0.5310 |
> > | Average Similarity | **0.6201** | **0.4462** | **0.6672** | **0.5975** | **0.4367** | **0.6756** | **0.5928** |  **0.4466** |  **0.6877** |
> > | Standard Deviation | 0.1357 | 0.1414 | 0.1367 | 0.1349 | 0.1336 | 0.1184 | 0.1358 | 0.1414 |  0.1333 |
> > | First Quartile | 0.5320 | 0.3581 | 0.5963 | 0.5147 | 0.3441 | 0.6036 | 0.5047 | 0.3540 | 0.6159 |
> > | Second Quartile | 0.6351 | 0.4468 | 0.6799 | 0.6023 | 0.4377 | 0.6881 | 0.5998 | 0.4492 |  0.6907 |
> > | Third Quartile | 0.7192 | 0.5495 | 0.7644 | 0.6906 | 0.5340 | 0.7621 | 0.6992 | 0.5329 | 0.7927 |

---

> > > ### Comment · Reviewer_XAPE · 2024-08-13
> > > **Thanks for the response.**
> > >
> > > Thanks for your response and experiment. However, I am a little confused (can't fully understand) about the setting in the response  experiment. I mean, maybe directly evaluating the similarity between the source text and the machine-generated text is enough to solve my concern?

---

> ### Author Response · Authors · 2024-08-13
> **Detailed Response by Author**
>
> Dear reviewer,
>
> your concern is whether the regenerated text **consistently** maintains a **high semantic similarity** with the original. We divided it into three aspects to address your concern:
>
> 1. **Semantic Similarity**
>
>     To evaluate the **semantic similarity** between the source text and the machine regenerated text, we measured the **cosine similarity** of their corresponding embeddings. Take the XSum dataset generated by ChatGPT (The First Column) as an example, the average cosine similarity is 0.6201.
>
> ---
>
> 2. **High Semantic Similarity**
>
>     There is an issue: given a value of cosine similarity, such as the previously mentioned 0.6201, does it indicate a high degree of similarity between the source text and the machine-generated text? To address this issue, we introduced a baseline similarity **for reference**, which better reflects the degree of semantic similarity reflected by a specific cosine similarity value.
>
>     The baseline similarity for reference is defined as the cosine similarity between human and AI texts for the same prompt. Since the texts correspond to the same prompt, their semantic similarity can be regarded as high. For example, in the XSum dataset generated by ChatGPT, the baseline similarity is 0.5042, which means that 0.5042 of cosine similarity already represents a high level of semantic similarity. The average similarity between the source text and the machine regenerated text is 0.6201, higher than 0.5042, meaning that they have **high** semantic similarity.
>
> ---
>
> 3. **Consistently Maintains High Semantic Similarity**
>
>     To evaluate whether the similarity between source text and the machine regenerated text **consistently** maintains a high semantic similarity. We measured the Standard Deviation, First Quartile, Second Quartile, and Third Quartile to better understand the distribution.
>
>     - Standard Deviation: A measure of how dispersed the data is in relation to the mean. In the XSum dataset generated by ChatGPT, the standard deviation is 0.1357, which is relatively small compared to the average of 0.6201, indicating **minimal fluctuation**.
>
>     Therefore, we can conclude that the regenerated text **consistently** maintains a high semantic similarity with the original.
>
> ---
>
> We greatly value every opportunity to discuss with you, **so we have conducted detailed experiments to address your concerns as thoroughly as possible.** I hope this explanation fully addresses your concern.

---

> > ### Comment · Reviewer_XAPE · 2024-08-13
> > **Official Comment by Reviewer XAPE**
> >
> > Thanks for the responses. I appreciate the author's detailed explanation, which addresses my concerns. I have revised my score accordingly.

---

> > > ### Author Response · Authors · 2024-08-13
> > > **Thank you for your response**
> > >
> > > We are delighted that our response addresses the reviewer's concerns. Again, we thank the reviewer for your valuable and positive feedback and comments.

---

### Official Review · Reviewer_qTwx · 2024-07-15

**Soundness:** 3
**Presentation:** 3
**Contribution:** 3
**Rating:** 6
**Confidence:** 4

**Summary:**

This paper develops an LLM-generated text detection method by ****. The key idea is to regenerate text based on a prompt reconstructed by an auxiliary LLM from the candidate text so that the regenerated text and the original candidate text can be used to extract similarity-based features, which are used for a classifier.

**Strengths:**

- (S1) The idea is simple and intuitive.
- (S2) Comprehensive experiments and analysis especially including paraphrased machine-generated text such as DIPPER and back-translate (in Table3)
- (S3) The paper is well-written and easy to follow.

**Weaknesses:**

- (W1) It is not clear if the proposed method can replace other classification-based methods. Ghostbuster [41] is known to be a strong machine-generated text detection method and should be compared for evaluation (it also offers a benchmark as well). Other recent simple-yet-effective methods could be compared as well. For example,
    - [Ref 1] Smaller Language Models are Better Zero-shot Machine-Generated Text Detectors https://aclanthology.org/2024.eacl-short.25.pdf
    - [Ref 2] Your Large Language Models Are Leaving Fingerprints https://arxiv.org/abs/2405.14057
- (W2) The datasets used for evaluation might be too easy to detect. Unless there’s a special reason, the proposed should also be tested in other benchmarks used for machine-generated text detection research (e.g., the Ghostbuster dataset).
- (W3) The feature extraction is computationally expensive. Two LLM inferences (prompt generation, candidate text regeneration) and then dual-encoder-based feature extraction. It is not clear if the feature extraction offers significantly better benefits compared to more lightweight approaches (e.g., [Ref 1][Ref 2] above)

### Minor comments

For perturbation, in addition to DIPPER, OUTFOX is another option.
    - [Ref 3] OUTFOX: LLM-Generated Essay Detection Through In-Context Learning with Adversarially Generated Examples https://arxiv.org/abs/2307.11729

**Questions:**

Please answer the weaknesses raised in the Weaknesses section.

**Limitations:**

Yes.

---

> ### Author Rebuttal · Authors · 2024-08-07
>
> We thank the reviewer for the time and expertise you have invested in these reviews. We are delighted to receive positive feedback that our work provides a solid contribution to the field, especially since the paper is well-written, the proposed idea is simple and intuitive, and the experiments and analysis are comprehensive.
>
> Below we provide point-by-point responses to your comments and questions.
>
> ---
>
> - **Weaknesses 1 & Weaknesses 2**: The datasets used for evaluation might be too easy to detect. Unless there’s a special reason, the proposed should also be tested in other benchmarks(e.g., the Ghostbuster dataset). & Add comparision with other classification-based methods, including Ghostbuster, Fingerprints,  Smaller Language Models.
>
>     **Response**: Following your advice, we compared detection performance on Ghostbuster benchmark[1]. As stated in Ghostbuster, out-of-domain performance yields the fairest comparison across training-based and zero-shot methods, so we mainly focus on the performance of out-of-domain generalization. In this benchmark, we compared with Ghostbuster[1], Fingerprints[2], and Smaller Models[3]. Additionally, we included comparisons with CoCo[4], a training-based method. We also included comparisons with RADAR and Fast-DetectGPT, which are the advanced training-based and zero-shot baselines in our paper. As the following Table shows,  the detection AUROC of Ghostbuster, Fast-DetectGPT and DPIC surpasses 0.98, indicating that Ghostbuster benchmark is easily detected by these methods.
>
>     |  | Methods | Out of domain |  |  |  |
>     | --- | --- | --- | --- | --- | --- |
>     |  |  | News | Creative Writing | Student Essays | Avg.  |
>     | Training-based | Ghostbuster | 0.9929 | 0.9839 | 0.9913 | 0.9893 |
>     |  | Fingerprints | 0.8108 | 0.8373 | 0.7846 | 0.8109 |
>     |  | CoCo | 0.9733 | **0.9979** | 0.9011 | 0.9574 |
>     |  | RADAR | **0.9984** | 0.8717 | 0.9418 | 0.9373 |
>     | Zero-shot | Smaller Models with Fast-DetectGPT (GPT-Neo-125M) | 0.9983 | 0.8957 | 0.9673 | 0.9537 |
>     |  | Fast-DetectGPT | 0.9979 | 0.9543 | **0.9965** | 0.9829 |
>     |  | DPIC | 0.9950 | 0.9978 | 0.9774 | **0.9900** |
>
>     Actually, our benchmark used in our original paper, already covers two domains from the Ghostbuster benchmark, namely Xsum for news articles and Writing for creative story writing. In addition, our benchmark also includes multiple prompts and three generative large language models, **offering a more rigorous evaluation standard.** Therefore, to better compare the performance of these newly added methods with DPIC, we further tested these methods using the benchmark employed in our paper.
>
>     We compared DPIC to these new detection methods in our benchmark. The results in **Table 1 of the attached PDF** show that DPIC performs best on average, which verifies its effectiveness. In detail, Ghostbuster only achieved an average detection AUROC of 0.8705, 0.9170, and 0.9239 in detecting different domain datasets generated by ChatGPT, GPT4, and Claude3, respectively, while DPIC demonstrates superior performance, achieving an average detection AUROC of 0.9634, 0.9734, and 0.9878, respectively. We will include this part in the final version of the paper.
>
>
> ---
>
> - **Weaknesses 3**: The feature extraction is computationally expensive. Two LLM inferences (prompt generation, candidate text regeneration) and then dual-encoder-based feature extraction. It is not clear if the feature extraction offers significantly better benefits compared to more lightweight approaches (e.g., [Ref 1][Ref 2] above)
>
>     **Response**: The feature extraction process in our method incurs certain costs, but the benefits it offers are significant. As Table 1 in our original paper shows, our method demonstrates superior generalization compared to existing approaches, achieving an average detection AUROC of 0.9806 in identifying texts from diverse domains generated by two commercial closed-source models: GPT4 and Claude3.
>
>     The superior performance brings practical value. Generally, a multi-stage detection strategy is practically effective for many detection tasks involving large amounts of generated text. Initially, a fast detection model with lower accuracy is used for preliminary screening. Once potentially generated text is identified, a more accurate detection model is employed in the second stage for detailed verification. Our method is well-suited as an accurate detection model for this second stage. Additionally, our method offers direct practical value in situations where time constraints are relaxed, and the volume of candidate texts is manageable, such as in academic plagiarism detection.
>
>
> ---
>
> - **Minor Comments**: For perturbation, in addition to DIPPER, OUTFOX is another option.
>
>     **Response**:  Good Suggestion. Following your advice, we added robustness detection experiments against OUTFOX attacks[5]. The results are shown in **Table 2 in the attached PDF**. Our method is somewhat affected, but its detection performance after the attack remains superior compared to the other methods. We will include this part in the final version of the paper.
>
> ---
>
> ### **References**
>
> Please see **Rebuttal References** in the global response.

---

> > ### Comment · Reviewer_qTwx · 2024-08-12
> >
> > Thank you for sharing the new experiment results and the responses. My concerns have not been addressed and I keep my initial evaluation as 6: Weak Accept.
> >
> >
> > >> (W1) & (W2)
> > >
> > > Actually, our benchmark used in our original paper, already covers two domains from the Ghostbuster benchmark, namely Xsum for news articles and Writing for creative story writing. In addition, our benchmark also includes multiple prompts and three generative large language models, offering a more rigorous evaluation standard. Therefore, to better compare the performance of these newly added methods with DPIC, we further tested these methods using the benchmark employed in our paper.
> >
> > Unless the Ghostbuster benchmark is a subset of the dataset, the Ghostbuster benchmark is considered another dataset and we cannot tell which one is better. The statement "offering a more rigorous evaluation standard." is not verified in the paper. Please clarify why the author(s) consider it.
> >
> >
> > >> (W3)
> >
> > This concern is not addressed. The feature extraction is effective with a certain cost.

---

> > > ### Author Response · Authors · 2024-08-12
> > > **Official Comment by Authors**
> > >
> > > Thank you again for the time and expertise you have invested in these reviews. We appreciate the opportunity to address the concerns and questions raised.
> > >
> > > - **(W1) & (W2):** Unless the Ghostbuster benchmark is a subset of the dataset, the Ghostbuster benchmark is considered another dataset and we cannot tell which one is better. The statement "offering a more rigorous evaluation standard." is not verified in the paper. Please clarify why the author(s) consider it.
> > >
> > >     **Response:** Dear reviewer, we sincerely apologize for comparing the benchmark merely through its domain, which is not full-considered. Since the Ghostbuster benchmark is not a subset of the our benchmark,  it is meaningful to evaluate detectors on the Ghostbuster benchmark as well.  Therefore, we will add the results on the Ghostbuster benchmark in the final version, and thanks again for your valuable suggestions.
> > >
> > >     And the statement 'offering a more rigorous evaluation standard' refers to the statement that the benchmark used in our paper includes three large language models. Since cross-model detection is a challenging problem in generated text detection[1,2],  we considered our benchmark is more rigorous in this aspect. From a strict perspective, we should not compare benchmark merely depending on the adopted LLMs. Therefore, we will add the results on the Ghostbuster benchmark in the final version, and not compare the benchmarks.
> > >
> > > ---
> > >
> > > - **(W3):** This concern is not addressed. The feature extraction is effective with a certain cost.
> > >
> > >     **Response:** Dear Reviewer, since we presented comparisons with the methods you mentioned in Weaknesses 1 & 2 of our rebuttal, we did not directly respond to the statement in Weakness 3: 'It is not clear if the feature extraction offers significantly better benefits compared to more lightweight approaches (e.g., [Ref 1][Ref 2] above).' We apologize for any confusion this may have caused and appreciate the opportunity to clarify further.
> > >
> > >     As shown in Table 1 of the attached PDF, we have compared our results with other methods, including the two lightweight methods you mentioned: Fingerprints [3], and Smaller Models [4]. To facilitate your review, we present the results below. Our method indeed shows a significant improvement over the lightweight methods in terms of average detection AUROC on both our benchmark and Ghostbuster benchmark.
> > >
> > >     I hope this explanation addresses your concern. If you have any further concerns or questions about our work, we are happy to discuss them with you. We will also add this part to strengthen the manuscript. Thank you again for the time and expertise you have invested in these reviews.
> > >
> > >     |  | ChatGPT |  |  |  | GPT4 |  |  |  | Claude3 |  |  |  | Ghostbuster |  |  |  |
> > >     | --- | --- | --- | --- | --- | --- | --- | --- | --- | --- | --- | --- | --- | --- | --- | --- | --- |
> > >     |  | XSum | Writing | PubMed | Avg. | XSum | Writing | PubMed | Avg. | XSum | Writing | PubMed | Avg. | News | Creative Writing | Student Essays | Avg. |
> > >     | Fingerprints | 0.8815 | 0.8073 | 0.6816 | 0.7901 | 0.8124 | 0.7896 | 0.7133 | 0.7718 | 0.8849 | 0.9033 | 0.8692 | 0.8858 | 0.8108 | 0.8373 | 0.7846 | 0.8109 |
> > >     | Smaller Models | 0.9835 | 0.9713 | 0.8843 | 0.9464 | 0.8818 | 0.9098 | 0.8234 | 0.8717 | 0.9798 | 0.9594 | 0.8868 | 0.9420 | **0.9983** | 0.8957 | 0.9673 | 0.9537 |
> > >     | DPIC | **1.0000** | **0.9821** | **0.9082** | **0.9634** | **0.9996** | **0.9768** | **0.9438** | **0.9734** | **1.0000** | **0.9950** | **0.9686** | **0.9879** | 0.9950 | **0.9978** | **0.9774** | **0.9900** |
> > >
> > > ## **References**
> > >
> > > [1] Bao G, Zhao Y, Teng Z, et al. Fast-DetectGPT: Efficient Zero-Shot Detection of Machine-Generated Text via Conditional Probability Curvature[C]//The Twelfth International Conference on Learning Representations.
> > >
> > > [2] Yang X, Cheng W, Wu Y, et al. DNA-GPT: Divergent N-Gram Analysis for Training-Free Detection of GPT-Generated Text[C]//The Twelfth International Conference on Learning Representations.
> > >
> > > [3] McGovern H, Stureborg R, Suhara Y, et al. Your Large Language Models Are Leaving Fingerprints[J]. arXiv preprint arXiv:2405.14057, 2024.
> > >
> > > [4] Mireshghallah N, Mattern J, Gao S, et al. Smaller Language Models are Better Zero-shot Machine-Generated Text Detectors[C]//Proceedings of the 18th Conference of the European Chapter of the Association for Computational Linguistics (Volume 2: Short Papers). 2024: 278-293.

---

### Author Rebuttal · Authors · 2024-08-07

### **Dear Reviewers,**
We appreciate the constructive and insightful comments from all the reviewers! We sincerely appreciate your time and effort in reviewing our work. We have provided detailed answers to the comments and questions from each reviewer in the different author responses. All the modifications will be represented in the final version.

Below are References and attached Tables and Figures mentioned in our response.

---

### **Rebuttal References**

[1] Verma V, Fleisig E, Tomlin N, et al. Ghostbuster: Detecting Text Ghostwritten by Large Language Models[C]//Proceedings of the 2024 Conference of the North American Chapter of the Association for Computational Linguistics: Human Language Technologies (Volume 1: Long Papers). 2024: 1702-1717.

[2] McGovern H, Stureborg R, Suhara Y, et al. Your Large Language Models Are Leaving Fingerprints[J]. arXiv preprint arXiv:2405.14057, 2024.

[3] Mireshghallah N, Mattern J, Gao S, et al. Smaller Language Models are Better Zero-shot Machine-Generated Text Detectors[C]//Proceedings of the 18th Conference of the European Chapter of the Association for Computational Linguistics (Volume 2: Short Papers). 2024: 278-293.

[4] Liu X, Zhang Z, Wang Y, et al. Coco: Coherence-enhanced machine-generated text detection under low resource with contrastive learning[C]//Proceedings of the 2023 Conference on Empirical Methods in Natural Language Processing. 2023: 16167-16188.

[5] Koike R, Kaneko M, Okazaki N. Outfox: Llm-generated essay detection through in-context learning with adversarially generated examples[C]//Proceedings of the AAAI Conference on Artificial Intelligence. 2024, 38(19): 21258-21266.

[6] Alibaba-NLP/gte-Qwen1.5-7B-instruct  (we are not allowed to post any link, so please google it to find the link)

[7] Wang Y, Mansurov J, Ivanov P, et al. M4: Multi-generator, Multi-domain, and Multi-lingual Black-Box Machine-Generated Text Detection[C]//Proceedings of the 18th Conference of the European Chapter of the Association for Computational Linguistics (Volume 1: Long Papers). 2024: 1369-1407.

[8] Qwen/Qwen1.5-7B-Chat (we are not allowed to post any link, so please google it to find the link)

[9] meta-llama/Meta-Llama-3.1-405B-Instruct (we are not allowed to post any link, so please google it to find the link)

---

### **Rebuttal Tables and Figures**

Please download the attached PDF file.

---

### Decision · Program_Chairs · 2024-09-25

**Decision:**

Accept (poster)

**Comment:**

This paper presents DPIC, a new framework for detecting LLM-generated text. The approach works by utilising an auxiliary LLM to "reconstruct the prompt of a candidate text and regenerate a text from it, allowing the detector to focus on the intrinsic characteristics of the generative model. The similarity between the candidate and regenerated texts is then used as a detection feature". The authors have included a number of additional experiments during the rebuttal, which have largely addressed the reviewers' concerns. The paper is well written and easy to follow. The idea presented is simple and it works and, in the words of one reviewer, demonstrates "robustness across various datasets, source models, and paraphrasing attacks, suggesting that the proposed method is capable of maintaining performance under different conditions and adversarial scenarios“. The primary issue with this work is that it seems rather incremental in nature, and is rather limited in terms of technical depth and novelty.